# Risk equations for prosthetic joint infections (PJIs) in UK: a retrospective study using the Clinical Practice Research Datalink (CPRD) AURUM and GOLD databases

Stefano Perni, Polina Prokopovich 

**Correspondence to**
Dr Polina Prokopovich;
prokopovichp@cf.ac.uk

## ABSTRACT

**Background** Prosthetic joint infections (PJIs) are a serious negative outcome of arthroplasty with incidence of about 1%. Risk of PJI could depend on local treatment policies and guidelines; no UK-specific risk scoring is currently available.

**Objective** To determine a risk quantification model for the development of PJI using electronic health records.

**Design** Records in Clinical Practice Research Datalink (CPRD) GOLD and AURUM of patients undergoing hip or knee arthroplasty between January 2007 and December 2014, with linkage to Hospital Episode Statistics and Office of National Statistics, were obtained. Cohorts' characteristics and risk equations through parametric models were developed and compared between the two databases. Pooled cohort risk equations were determined for the UK population and simplified through stepwise selection.

**Results** After applying the inclusion/exclusion criteria, 174 905 joints (1021 developed PJI) were identified in CPRD AURUM and 48 419 joints (228 developed PJI) in CPRD GOLD. Patients undergoing hip or knee arthroplasty in both databases exhibited different sociodemographic characteristics and medical/drug history. However, the quantification of the impact of such covariates (coefficients of parametric models fitted to the survival curves) on the risk of PJI between the two cohorts was not statistically significant. The log-normal model fitted to the pooled cohorts after stepwise selection had a C-statistic >0.7.

**Conclusions** The risk prediction tool developed here could help prevent PJI through identifying modifiable risk factors pre-surgery and identifying the patients most likely to benefit from close monitoring/preventive actions. As derived from the UK population, such tool will help the National Health Service reduce the impact of PJI on its resources and patient lives.

## STRENGTHS AND LIMITATIONS OF THIS STUDY

⇒ We sampled a large cohort of patients and considered a wide range of covariates and of drugs to establish the medical and medication history.

⇒ The specific geographical location (UK) of the patients analysed may limit the availability of data regarding the impact of ethnicity and not have the same predictive power when applied to other countries.

⇒ It is plausible some confounding variables were unavailable, either because they are not routinely recorded (for example, nutritional status and genetic factors) or unavailable (for example, hospital prescribing and use of over-the-counter medications).

⇒ The use of registry-based data incurs the possibility of errors in the recording of data, or codes may be a source of information bias.

⇒ Our study was observational thus only able to identify correlations between prosthetic joint infections and the included covariates, not causality.

## INTRODUCTION

Infections associated with arthroplasty, known as prosthetic joint infections (PJIs), have an incidence of about 1% after primary arthroplasty; this is about 5 times higher after secondary arthroplasty not resulting from a previous PJI (aseptic replacement) while about 10 times after a revision caused by a PJI.[1 2] The consequences of developing a PJI can be extremely debilitating in the long term for patients,[3] and with an estimate mean cost to the National Health Service (NHS) of ~£30 000 for PJI in knee joints[4] and ~£24 000 for PJI in hips; thus, healthcare providers also incur high costs.

Numerous risk factors for PJI have been identified and risk quantification tools developed[5–9] as the possibility of identifying patients at risk of PJI would allow a more proactive and targeted approach to prevention; however, the quantification of an individual probability to develop PJI is often impacted by the size of the cohort used in deriving such risk equations and the covariates available in the datasets employed for such analysis.[10 11] Such limitations reduce

the ability to translate conclusions to other settings/ countries and limit the accuracy of the estimated risk for patients.

Moreover, UK-specific risk and contemporary equations have not been developed yet and such PJI risk estimates are calculated based on results from different countries (such as Sweden)[5] where treatment algorithms and local medical guidelines may differ from the UK.

Clinical Practice Research Datalink (CPRD) is a UK government not-for-profit research service; it collects health records from participating general practitioner (GP) practices that have agreed to contribute data to this database related to diagnoses, symptoms, prescriptions, referrals and tests for patients. It has provided anonymised primary care data to conduct public health research.[12] CPRD GOLD and AURUM databases have been shown to be a data source with quality and completeness.[13] CPRD also provides linkage to the Office of National Statistics (ONS), providing accurate date and cause of death, along with Hospital Episode Statistics (HES) that contains records of secondary care activities including dates, specialty, clinical diagnoses and procedures.[12] Due to some differences in the structure and clinical coding of information technology systems employed by the GP practices,[14] CPRD data are provided as separate databases: CPRD GOLD contains data contributed by practices using Vision software,[15] while CPRD AURUM contains data from practices using EMIS Web.[12] In September 2018, CPRD AURUM included records of 19 million patients registered in 738 practices (~10% of all English practices), of whom 7 million were alive and actively contributing (13% of the population of England). In 2013, CPRD GOLD contained records from 11 million patients registered in 674 practices, including 4.4 million patients active (alive and currently registered), approximately 6.9% of the UK population.[15] Therefore, they both offer a representative population of UK patients; however, when linkage to HES and ONS is employed, only patients registered in GP practices located in England are represented in the final dataset.

Because of the lack of contemporary and UK-specific risk equations for the development of PJI, the objectives of this study were two-folds. First, we wanted to compare the characteristics of patients developing PJI in both CPRD databases, then to estimate risk equations from each database and compare the contribution of each individual covariate in both databases to assess the possibility of pooling participants. Second, relevant risk factors were identified from a stepwise regression of the pooled cohorts' risk equation and the predictive ability of the newly developed risk equation assessed.

## METHODS
### Data sources
Data were obtained from both databases available in the CPRD. Furthermore, CPRD data were linked to HES secondary care data and ONS mortality data.

### Study design and population
This was a retrospective study of patients undergoing hip or knee arthroplasty; the index date for inclusion was the day of arthroplasty. Patients had to be aged ≥30 years at index date and have arthroplasty between 1 January 2007 and 31 December 2014. Patients were excluded from the study if they were registered on the CPRD for <6 months prior to index date with the most recent CPRD up-to-standard date >6 months prior to index date.

This research is largely descriptive rather than inferential in nature in that it aims to characterise patterns of PJI and verify the generalisability of existing risk equations. A sample size and power calculation have, therefore, not been undertaken with respect to identifying differences in outcomes between groups.

Eligible patients were identified through Office of Population Censuses and Surveys (OPCS-4) Classification of Interventions and Procedures codes[2] for hip or knee arthroplasty in HES records and they were followed up for 5 years or the first occurrence of: device replaced, death or loss to follow-up. Loss to follow-up was defined as the date a patient was transferred out of the practice, the date that the practice left the database or the last recorded event date from CPRD extraction.

### Study measures
Covariates related to patient characteristics were extracted from the CPRD database; surgery properties were derived from the HES database according to OPCS codes (for type of fixation, laterality fixation, grafts) and admission codes. Medical history was determined by the presence before the index date of disease-specific ICD-10 (10th revision of the International Statistical Classification of Diseases and Related Health Problems) codes in HES or either MedCodeID codes in the CPRD AURUM database or READ codes in CPRD GOLD database. Prescription history was assessed from relevant British National Formulary codes reported in the CPRD database.[2] Data cleaning was performed to remove implausible entries such as day of death before index date and body mass index (BMI) <10 or BMI >80.

PJI occurrence was assessed by the presence of the ICD-10 code T84.5 in the HES database; the date of PJI occurrence was set as the day of hospitalisation. To ascertain that PJIs diagnosed were affecting the joint of interest as patients could have multiple devices, only a diagnosis of PJI and a record to OPCS of any procedure in the joint of interest during the hospitalisation were considered.

### Missing data handling
Medical history was based on the presence of a specific diagnostic code; therefore, the absence of such codes in a patient's records was assumed to represent absence of such event, as such, no missing data were possible in a patient's medical history. On the contrary, when a record was expected, such as in the case of BMI, age,

primary or secondary arthroplasty along with alcohol and smoking status, the absence of any entry resulted in this patient having the covariate under consideration categorised as missing. Missing data were handled through categorical binning and no imputation methods applied.

## Survival analysis

Survival curves for each database were fitted with parametric models (exponential, Weibull, normal, lognormal, logistic, log-logistic) and ranked according to Akaike Information Criteria (AIC) and Bayesian Information Criteria (BIC).

Stepwise regression was employed to reduce the number of covariates in the risk equation using AIC as criteria to identify which variable to add/remove or to stop the selection process.

## Statistical analysis

Descriptive analyses were generated, characterising patient demographics, clinical and treatment characteristics. Summary statistics (eg, mean, SD, median, IQR, minimum and maximum) were calculated for continuous variables (age and BMI), while number and proportion/percentage were calculated for categorical variables. For multivariate regression, age was categorised as ≤45, 46–55, 56–65, 66–75, 76–85 and >85 years, while BMI was categorised as ≤20, 20–25, 26–30, 31–35, 36–50 and >50 years. The number and proportion of patients with missing data are also determined for each of the variables of interest.

Kaplan-Meier curves were generated, and the log-rank test was used to compare survival curves between the CPRD AURUM and CPRD GOLD databases.

PJI rates were calculated dividing the number of PJI cases observed by the total duration of the follow-up.

Z-test for each factor was performed to compare the regression coefficients between risk equations developed using CPRD AURUM and CPRD GOLD.[16]

$$Z_i = \frac{\beta_{i,AURUM} - \beta_{i,GOLD}}{\sqrt{SE_{i,AURUM}^2 + SE_{i,GOLD}^2}}$$

where:
$\beta_i$=regression coefficient for factor $i$.
$SE_i$=SE of regression coefficient for factor $i$.

The predictive ability of the final risk equation was assessed by calculating the area under the precision–recall and receiver operating curves at time-varying follow-ups.

C-statistic was evaluated after different follow-ups; 95% CIs were estimated through bootstrapping with 500 replicates for each time point.

Data collection, analysis and visualisation were performed using R (V.4.2.2) and relevant packages.[17 18]

## Patient and public involvement

Patients and/or the public were not involved in the design, or conduct, or reporting, or dissemination plans of this research.

# RESULTS

## Patient counts

After linkage with both HES and ONS, records for 330173 hip or knee arthroplasty, corresponding to 235249 patients, were identified in CPRD AURUM; similarly, CPRD GOLD contained information for 88458 hip or knee joints subjected to arthroplasty related to 62672 patients. After applying the inclusion/exclusion criteria, the records of 174905 joints (144048 patients) were identified in CPRD AURUM and 48419 joints (39764 patients) in CPRD GOLD (online supplemental table 1).

In patients indexed in either CPRD AURUM or GOLD, the most common reason for not completing the study observation period (5 years) was death, 14.67% and 12.44% in CPRD AURUM and GOLD, respectively. The second most common reason in CPRD AURUM was transfer out of the CPRD database (in 12.91% of cases), while in CPRD GOLD, it was data collection terminated before end of the study (in 34.29% of cases); replacement of the implanted device in the first 5 years after surgery occurred in about 2.3% of cases in both databases. A diagnosis of PJI in the first years after implant occurred in 1021 joints (0.58%) in CPRD AURUM and 228 joints (0.47%) in CPRD GOLD (online supplemental table 2).

Patients not completing the study observation period were followed up for a median of 726 and 768 days in CPRD AURUM and GOLD, respectively (online supplemental table 3).

## Baseline characteristics

The demographic and baseline characteristics of patients in both CPRD databases were generally similar for all covariates tested (eg, gender, age, BMI, alcohol and smoking status) (online supplemental table 4); the level of unknown BMI was greater in AURUM than in GOLD. The characteristics of the arthroplasty exhibited similar distribution in both databases (online supplemental table 6).

Patients undergoing hip or knee arthroplasty had similar prevalence of most of the diseases included in this study; however, chronic kidney disease, rheumatoid arthritis, myocardial infarction (MI), anaemia and cancer had different distributions in AURUM and GOLD (online supplemental table 6). Moreover, the history of prescriptions in patients with hip and knee arthroplasty in CPRD AURUM and GOLD was comparable (online supplemental table 7).

## PJI occurrence

Unadjusted Kaplan-Meier curve for time to PJI diagnosis showed a decreasing hazard rate with time; 0.21% and 0.32% of joints in CPRD AURUM developed PJI in the first 6 and 12 months, respectively. After 2 and 4 years, the proportion of joints with PJI was 0.45% and 0.61%, respectively. The probability of developing PJI in the joints from CPRD GOLD was 0.15%, 0.25%, 0.42% and 0.56% after 6 months, 1 year, 2 years and 4 years, respectively. The log-rank test between the two curves had p<0.001 (figure 1A).

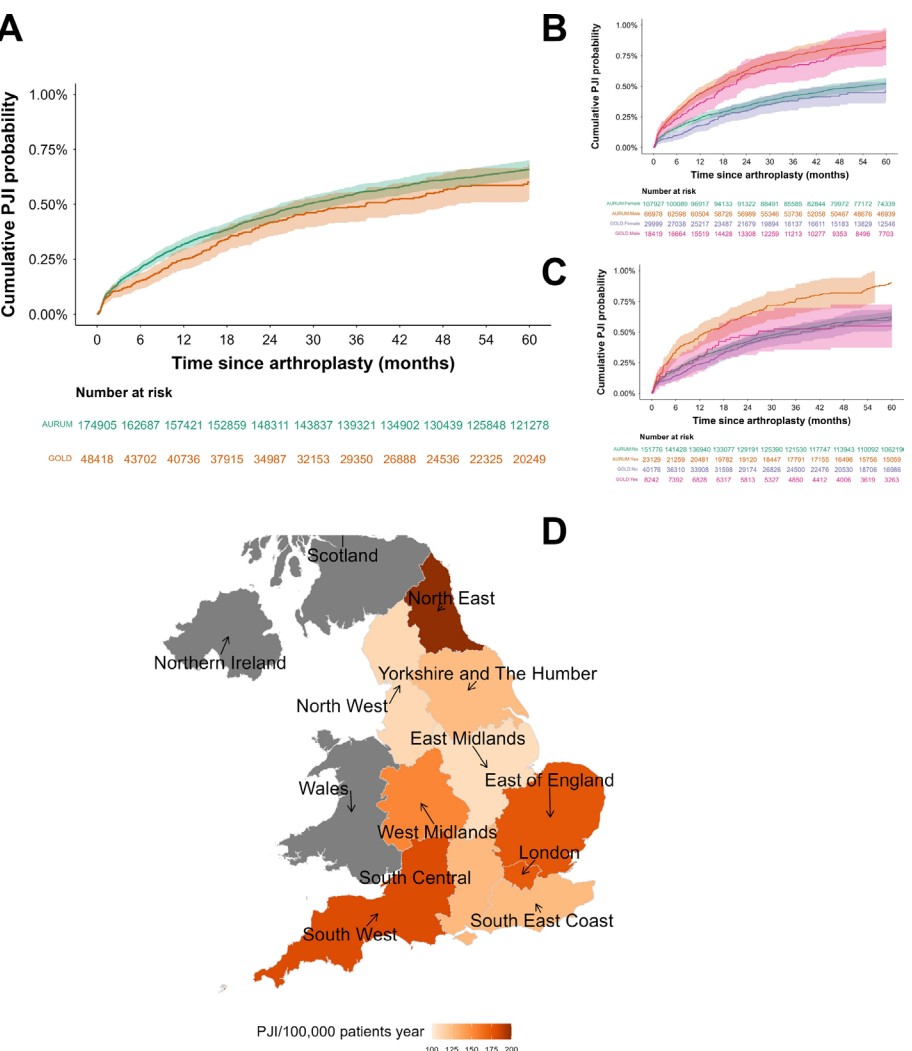

**Figure 1** Examples of unadjusted Kaplan-Meier curves of cumulative risk of PJI (and 95% CIs) for the entire cohorts (A) and stratified based on gender (B) and history of diabetes at baseline (C). Geographical distribution of PJI rates in the CPRD regions of England (D). CPRD, Clinical Practice Research Datalink; PJI, prosthetic joint infection.

Likewise, the Kaplan-Meier curves for time to PJI in the two CPRD databases exhibited comparable patterns with p values generally <0.001 (figure 1B,C).

The distribution of the PJI rate across the different England regions defined in CPRD ranged from ~100 PJI cases/100 000 patient-years in the East Midlands to ~180 PJI cases/100 000 patient-years in the North East (figure 1D).

### Risk equations

Survival curves of time to PJI diagnosis in CPRD GOLD and CPRD AURUM were fitted independently with parametric models; for both databases, the log-normal distribution returned the best fitting (eg, lowest AIC, lowest BIC and highest log-likelihood) (online supplemental tables 8 and 9). The coefficients of the log-normal parametric models for both databases were compared with Z-test and generally had p>0.05 (figure 2 and online supplemental table 10).

The risk equations based on the log-normal survival model developed using the pooled cohorts ability to

categorise patients as at risk of PJI or not over different follow-up times increased with longer periods as revealed by the area under the precision–recall curve; the area under the receiver operating curve and C-statistic did not exhibit the same pattern of increasing performance with longer follow-up (figure 3 and online supplemental table 11).

The stepwise variable selection was employed to reduce the number of covariates in the risk equation to those most significant. The final list of coefficients revealed that gender, age and BMI were patient characteristics that represent risk factors for PJI. The type of surgery (primary or secondary), admission through Accident and Emergency or elective and resurfacing of the patella were the risk factors related to the surgical procedure. Having a history of atrial fibrillation, deep vein thrombosis, diabetes, osteoarthritis, MI and hypertension increased the risk of PJI. Previous PJIs, even in an unrelated joint, were also medical risk factors for a further PJI. The use of non-steroidal anti-inflammatory drugs (NSAIDs) and

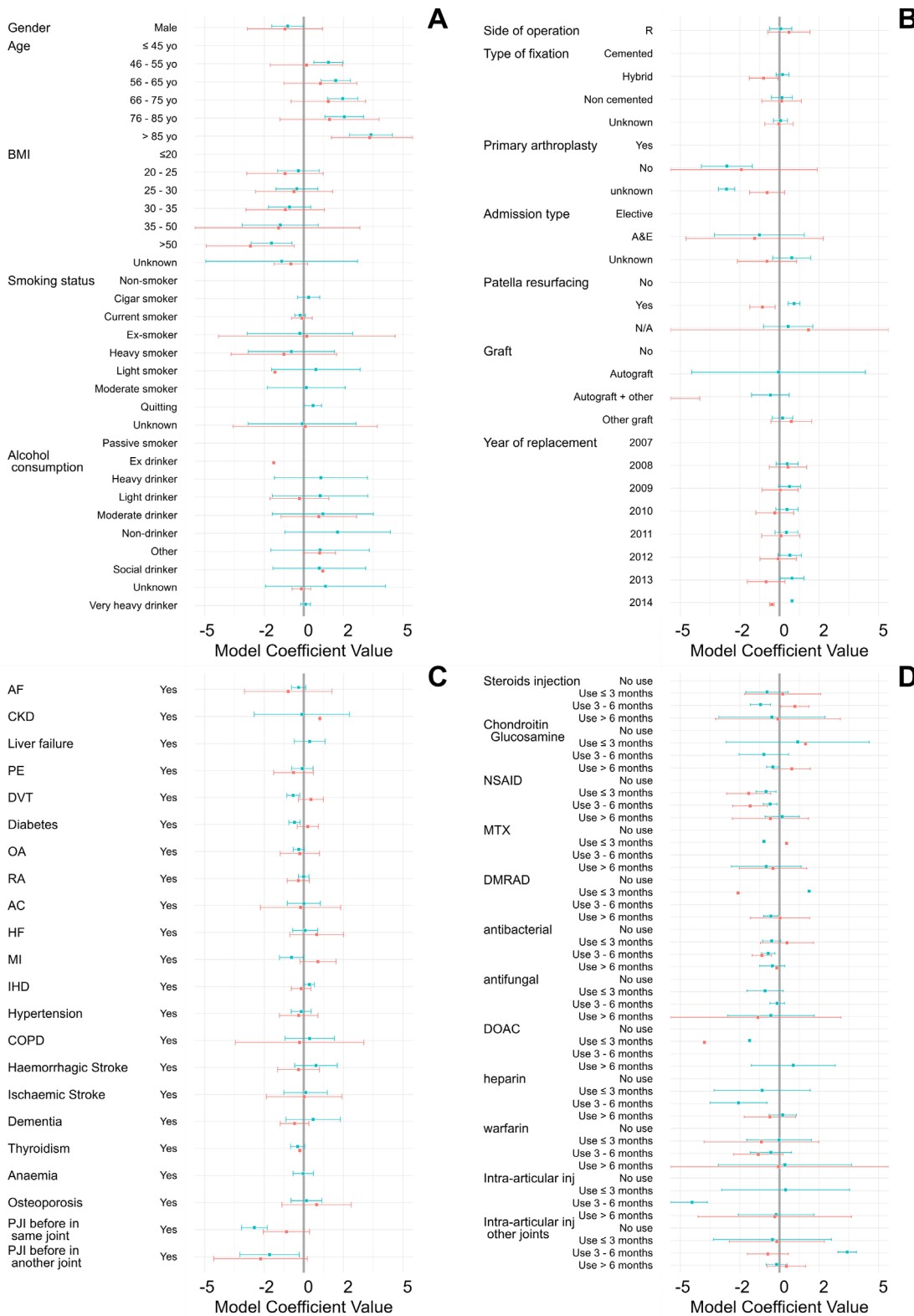

**Figure 2** Coefficients for time to PJI multivariate regression with log-normal model for data from CPRD GOLD (orange lines) and AURUM database (green lines). (A) Patients characteristics, (B) Surgery characteristics, (C) Medical history and (D) Medication history. A&E, Accident and Emergency; AC, active cancer; AF, atrial fibrillation; BMI, body mass index; CKD, chronic kidney disease; COPD, chronic obstructive pulmonary disease; CPRD, Clinical Practice Research Datalink; DMRAD, disease-modifying rheumatoid arthritis drug; DOAC, direct oral anticoagulant; DVT, deep vein thrombosis; HF, heart failure; IHD, ischaemic heart disease; MI, myocardial infarction; MTX, methotrexate; N/A, not applicable; NSAID, non-steroidal anti-inflammatory drug; OA, osteoarthritis; PE, pulmonary embolism; PJI, prosthetic joint infection; R, right; RA, rheumatoid arthritis.

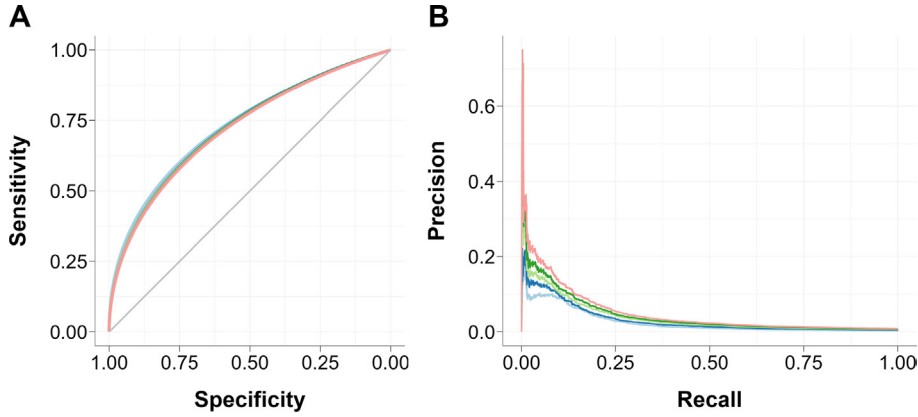

**Figure 3** Time-dependent (light blue line: 1 year, dark blue line: 2 years, light green line: 3 years, dark green line: 4 years, red line: 5 years) receiver operating curve (A) and precision–recall curve (B) of pooled log-normal equation.

antifungal drugs before arthroplasty was also a risk factor (table 1). Examples of the fitting of the Kaplan-Meier curves with the log-normal stepwise regression model showed a generally good agreement (figure 4).

## DISCUSSION

PJI is a relatively rare but highly impactful outcome, both in terms of patients' quality of life and resource consumption,[3 4 19–22] of arthroplasty. Prevention or early identification of PJI through stringent medical follow-up is not a viable option because of the large population undergoing hip and knee arthroplasty and the potential long follow-up time; as shown here and in multiple studies, PJI can be diagnosed even years after surgery.[2 23 24] Because of the long follow-up required to capture the full extent of PJI, we chose a study period of 5 years, and to allow patients to achieve such observation, we restricted the last date of index surgery to 5 years before the last date of collection in CPRD extraction analysed. In light of the administrative nature of the CPRD database, medical history was based on the presence of a specific diagnostic code; therefore, the absence of such code in a patient's records was assumed to represent absence of such event. On the contrary, when a record was expected, such as in the case of BMI, primary or secondary arthroplasty along with alcohol and smoking status, the absence of any entry resulted in the patient having the covariate under consideration categorised as 'unknown'.

Patients identified in CPRD AURUM were statistically different than those identified in CPRD GOLD (online supplemental tables 4–7) likely as a consequence of the fact that different geographical areas of England covered by the two databases were associated with the variation of population characteristics around the country in terms of age and other diseases prevalence/incidence.[25 26] However, as expected, the impact of almost all covariates on the risk of developing PJI was the same (p>0.05) when estimated with the same parametric model. The hazard of PJI (figure 1) exhibited a monotonically decreasing profile typical of other studies investigating such an occurrence,[20 27–30] and in both databases, the log-normal

model returned the best fitting of the data as this model is suitable for this type of risk profile.[31] Similarly, the log-logistic, another model suitable for decreasing hazard,[32] also returned good fitting (online supplemental tables 8 and 9).

The observed PJI incidence in this study (~0.7% after 5 years) is lower than that sometimes reported. One possible explanation for this was the use of a specific ICD-10 code for the identification of PJI (T84.5), while in our studies that reported higher PJI rates (about 2.5% over 5 years), additional ICD-10 codes were employed to identify PJI, for example, T81.4 (infection following a procedure) and T84.7 (infection and inflammatory reaction due to other internal orthopaedic prosthetic devices, implants and grafts)[33]; however, such codes appear less specific as, for example, T81.4 may lead to considering a superficial infection of the surgical wound as a PJI, while M00 corresponds to a diagnosis of pyogenic arthritis.[5] When using only the presence of ICD-10 code T84.5, the PJI incidence at 5 years observed in this study is similar to the 0.89% previously reported.[34]

As regression coefficients in the risk equations obtained from both databases were not statistically different for all covariates included in this study, both cohorts were pooled to increase the population size and, consequently, reduce the error associated with the coefficient estimation.[35 36] The regression model built using the pooled cohort performance was evaluated using the area under the receiver operating curve that represents the balance between sensitivity (correctly identified cases) and specificity (correctly identified non-cases). The model performance did not significantly vary when applied to the likelihood of developing PJI over different follow-up times, probably due to the unbalance nature of the cohort in terms of patients developing PJI and patients not developing PJI. The area under the precision–recall curve (figure 3B) instead is not impacted by unbalanced samples and the model performed better with increasing follow-up time.[37]

Stepwise selection is a classic variable reduction technique and was employed here to simplify the model

**Table 1** Coefficients and SE of pooled log-normal parametric model of time to PJI after stepwise regression

| Variable | Coefficient | SE |
|---|---|---|
| Log(scale) | 1.644 | 0.025 |
| Intercept | 20.724 | 0.657 |
| Male | −0.827 | 0.111 |
| Age | | |
| ≤45 | Reference | |
| 46–55 | 1.036 | 0.37 |
| 56–65 | 1.469 | 0.343 |
| 66–75 | 1.823 | 0.341 |
| 76–85 | 1.917 | 0.349 |
| >85 | 3.388 | 0.452 |
| BMI | | |
| ≤20 | Reference | |
| 20–25 | −0.462 | 0.477 |
| 26–30 | −0.424 | 0.467 |
| 31–35 | −0.812 | 0.471 |
| 36–50 | −1.257 | 0.478 |
| >50 | −1.803 | 0.886 |
| Unknown | −1.15 | 0.461 |
| Primary arthroplasty | | |
| Yes | Reference | |
| No | −2.558 | 0.158 |
| Unknown | −2.467 | 0.606 |
| Admission type | | |
| Elective | Reference | |
| A&E | −1.064 | 0.185 |
| Unknown | 0.295 | 0.976 |
| Patella resurfacing | | |
| No | Reference | |
| Yes | 0.393 | 0.409 |
| N/A | 0.714 | 0.118 |
| AF | −0.484 | 0.193 |
| DVT | −0.392 | 0.23 |
| Diabetes | −0.357 | 0.148 |
| OA | −0.237 | 0.125 |
| MI | −0.329 | 0.182 |
| Hypertension | −0.312 | 0.105 |
| PJI before in same joint | −2.585 | 0.302 |
| PJI before in another joint | −1.818 | 0.257 |
| NSAID | | |
| No use | Reference | |
| Use <3 months | −0.768 | 0.138 |
| Use 3–6 months | −0.589 | 0.22 |
| Use >6 months | 0.112 | 0.14 |
| Antifungal | | |

Continued

**Table 1** Continued

| Variable | Coefficient | SE |
|---|---|---|
| No use | Reference | |
| Use <3 months | −0.725 | 0.315 |
| Use 3–6 months | −0.096 | 0.457 |
| Use >6 months | −0.585 | 0.161 |

A&E, Accident and Emergency; AF, atrial fibrillation; BMI, body mass index; DVT, deep vein thrombosis; MI, myocardial infarction; N/A, not applicable; NSAID, non-steroidal anti-inflammatory drug; OA, osteoarthritis; PJI, prosthetic joint infection; SE, Standard error.

minimising the loss of performance. Negative coefficients indicate an increased risk of PJI compared with the reference; analogously, positive coefficients indicate a reduced risk of PJI compared with the reference. The greater the absolute value of the coefficient, the greater the impact of the risk of PJI. Most of the risk factors identified through this approach are in line with those commonly associated with PJI (table 1) as male gender, increased BMI and a previous PJI in the same joint or in another joint were retained in the list of covariates after stepwise selection.[2 9 29 38] Age is sometimes not reported as a risk factor for PJI[9 39]; however, the protective role of ageing in PJI observed in this study was also seen previously.[34] A possible explanation for such correlation is 'survival bias' as older patients dying at higher rates would be censored at the time of death before having the opportunity of developing PJI. Smoking is sometimes reported as a risk factor of PJI[9 40] because it causes delay in wound healing but sometimes the opposite is reported,[41] such as in this study, where patients' smoking status was not included in the final model. Moreover, the active role of numerous medications provides a range of modifiable risk factors that could also inform surgery scheduling when arthroplasty is an elective procedure as the risk of PJI after NSAID and antifungal use decreases with increasing interval between the last use and arthroplasty. The plausibility of the role of NSAIDs in PJI could be linked to the increased cardiovascular risk resulting from their use[42] that in turn increases the risk of PJI; it is also widely accepted that immunosuppression is a general risk factor for the development of PJI and as such, NSAIDs' role in PJI can also be linked to the immunological activity.[43] The models did not identify the use of steroidal injection in the joint as a risk associated with PJI in line with the advice of the American Academy of Orthopedic Surgeons.[44] The use of antifungal drug prior to arthroplasty has not been generally investigated as a risk factor for PJI; it is possible that a prescription of antifungal drugs before the joint replacement surgery is a risk factor in our model as this would indicate a patient more susceptible to infections.

Compared with other available models, besides the UK-specific population, our model includes patients both with high BMI (generally considered more at risk

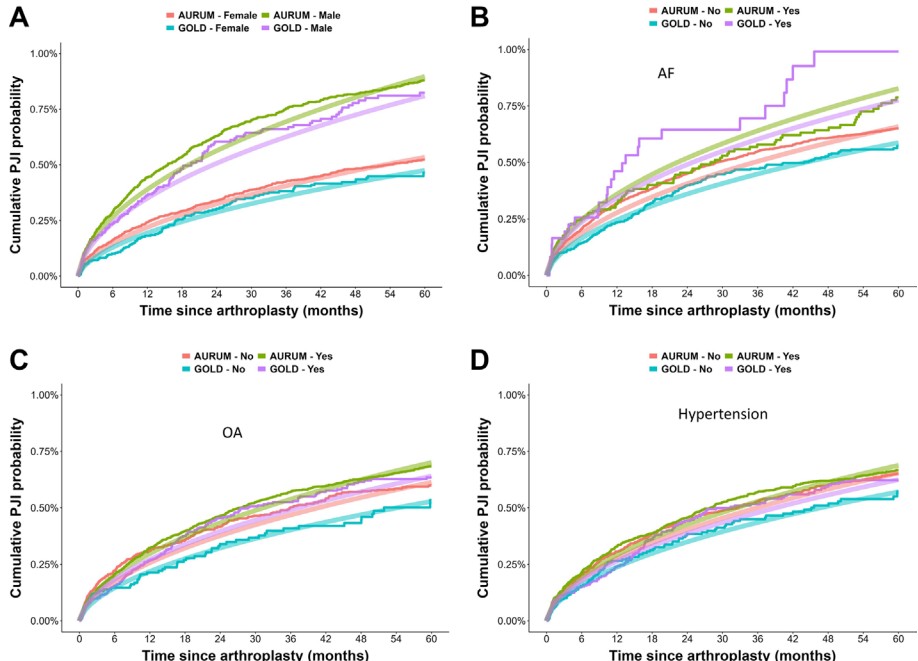

**Figure 4** Examples of Kaplan-Meier curves of risk of PJI and prediction using pooled log-normal parametric model after stepwise selection for both AURUM and GOLD CPRD databases: (A) gender and history of (B) atrial fibrillation (AF), (C) osteoarthritis (OA) or (D) hypertension at baseline. CPRD, Clinical Practice Research Datalink; PJI, prosthetic joint infection.

of PJI) or unknown BMI; a previous risk scoring model excluded patients with either missing BMI information or BMI >50.[5] The model developed here considered a wider range of covariates compared with the Mayo Prosthetic Joint Infection Risk Score[7] that included only diabetes or a limited number.[45] Moreover, the model presented here is not specific for any pathogen in contrast with a *Staphylococcus aureus*-specific model[46]; this particular relevance was in light of the multitude of pathogens isolated in PJI such as those that were Gram positive and Gram negative, and fungi.[2] Additionally, the risk equation presented here has also consideration of patients' drug history that was very limited, if not missing at all, in other published models.[5 7 45 46]

The model performance after stepwise selection was still high as shown by the C-statistic (online supplemental table 11) and the visual fit of the curves (figure 4) demonstrating how the retained covariates are pivotal in determining a patient's risk of developing PJI after arthroplasty.

The model developed here does not simply provide a list of risk factors for PJI but enables an objective quantification of the risk to develop PJI after hip or knee arthroplasty. As most of the identified risks are not modifiable, the application of this risk scoring algorithm would not primarily allow a personalised pre-surgery drug prescription consideration but will guide follow-up care pivoting the resources (visits, test, etc) to the subgroup of patients at higher risk of PJI.

As in most retrospective studies, there were numerous missing data. Missing data have been handled through categorical binning and no mechanisms related to missingness have been hypothesised (ie, missing at random or missing not at random). In order to avoid selection bias,

the analysis has not been restricted to complete cases; however, the criteria requiring data at baseline for certain variables may invite bias, as the patient profile of those with complete data may be different to those without. Nevertheless, a workable dataset is required to fulfil the study objectives and, therefore, this practical approach was adopted to identify a suitable study population. Another possible limitation is that comorbidities were considered as binary (present or absent), without accounting for illness severity and only at baseline without accounting for time-varying factors. Despite these acknowledged limitations, the large population size, the quality of the data in the database and the numerous covariates considered are considerable strengths of this work. Sensitivity analysis and multiple imputations could be further applied to refine the risk scoring algorithm presented.

## CONCLUSIONS

Patients undergoing hip or knee arthroplasty present in CPRD AURUM and CPRD GOLD exhibit different sociodemographic characteristics and medical/drug history. However, the coefficients of parametric models, fitted to the survival curves, were not statistically different, allowing for pooling cohorts.

The model developed here demonstrated a good ability to identify patients at risk (based on C-statistic) and is based on the UK-specific population. Such capability would allow the NHS to review the current guidelines, potentially devising a targeted programme of medical treatment pre-surgery, when appropriate, and of close monitoring of a small fraction of patients

after undergoing hip or knee arthroplasty, allowing to reduce the incidence and impact of PJI.

The newly developed risk equations will also provide contemporary inputs for the economic evaluation of new technologies for the prevention of PJI, supporting Health Technology Assessment from the National Institute for Health and Care Excellence.

**Contributors** SP and PP conceived the idea for the study. SP carried out the statistical analysis and wrote the first draft of the manuscript. PP supervised the study and analysis. All authors reviewed and revised the manuscript. PP acts as the guarantor.

**Funding** This work has been supported by Wellcome Trust (WT109455MA).

**Disclaimer** The funder had no role in the study design, data collection and analysis, decision to publish or preparation of the manuscript. The interpretation and conclusions contained in this study are those of the author/s alone.

**Map disclaimer** The inclusion of any map (including the depiction of any boundaries therein), or of any geographic or locational reference, does not imply the expression of any opinion whatsoever on the part of BMJ concerning the legal status of any country, territory, jurisdiction or area or of its authorities. Any such expression remains solely that of the relevant source and is not endorsed by BMJ. Maps are provided without any warranty of any kind, either express or implied.

**Competing interests** None declared.

**Patient and public involvement** Patients and/or the public were not involved in the design, or conduct, or reporting, or dissemination plans of this research.

**Patient consent for publication** Not applicable.

**Ethics approval** This study protocol (19_009) was reviewed by the Independent Scientific Advisory Committee (ISAC) and approved in January 2020; a minor amendment was subsequently approved on 1 December 2021. Data were extracted in December 2020. The research team had access to only de-identified data; therefore, the requirement for patient consent was waived.

**Provenance and peer review** Not commissioned; externally peer reviewed.

**Data availability statement** Data may be obtained from a third party and are not publicly available. This study is based in part on data from the Clinical Practice Research Datalink obtained under license from the UK Medicines and Healthcare products Regulatory Agency. The data are provided by patients and collected by the NHS as part of their care and support. The authors did not have any special access privileges that others would not have. Other researchers can apply for access to the raw data at the following: https://cprd.com/data-access/.

**ORCID iD**
Polina Prokopovich http://orcid.org/0000-0002-5700-9570

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
