## [Reviewer comments · BMJ Open]

ARTICLE DETAILS

TITLE (PROVISIONAL)	Retrospective study in UK using the Clinical Practice Research Datalink (CPRD) AURUM and GOLD database for the determination of risk equations for prosthetic joint infections (PJI)
AUTHORS	Perni, Stefano; Prokopovich, Polina

VERSION 1 – REVIEW

REVIEWER	Sandra Nelson Massachusetts General Hospital
REVIEW RETURNED	18-Jan-2024

GENERAL COMMENTS	For the editor: I did not realize when I accepted this review that the needs of the reviewer would be very heavily weighted towards an understanding of tool development and validation. These are not within my skill set. While I am able to comment on how this study fits into the available literature, I am not sufficiently able to evaluate it on methodologic grounds, and would certainly ensure that there is appropriate evaluation from a statistician or methodologist. Additionally, please note that I am a US physician and my understanding of care systems in the UK is more limited. General Comments In this study, the authors sought to develop a risk prediction tool for the development of periprosthetic joint infection following arthroplasty. Several such tools are in existence already but the available tools have primarily been developed in non-UK populations. There are important differences in both population comorbidity risk as well as surgical methods such that development of a tool that is more directly relevant to the UK population is a very reasonable endeavor. The authors used two publicly available data sets to develop the risk prediction tool. Despite differences in the available data sets, the authors were able to demonstrate the reproducibility of the prediction tool which may lead to targeted risk mitigation strategies. Important issues 1. The reported PJI risk identified in both databases was low (0.4% and 0.5%). For a dataset that is meant to include both early and late PJI, I question whether there may have been missed outcomes in the dataset, as most studies put the true incidence as higher. The authors should address this in their discussion, and whether or not this might introduce bias.2. The authors included a cohort who underwent their index procedure as long ago as 2007. There have been important improvements in preventive measures over this time that reduced PJI risk (e.g. Staph aureus testing and decolonization). The authors should consider including the time of the index procedure as a covariate.
---

	3. The authors extracted data in December 2020 (methods line 115) and included individuals who underwent index arthroplasty as late as July 2020, yet they also stated that patients were followed up for 5 years or until the first occurrence of the outcome (methods line 130). These are internally inconsistent, and according to table S2, a significant minority of persons in both databases did not complete the 5 year followup. While the data was censored for these individuals, this may still bias the prediction tool towards risk factors associated with early PJI. The authors might consider separate prediction tools based on time following arthroplasty (e.g. early risk – within one year, and later risk, beyond one year after index procedure) or consider this as a sensitivity analysis. If this is not done, the authors should address this in the discussion section. 4. The discussion does not really address how this study fits into the available literature. Much of it reads more to me like a supplemental methods section. I would like the authors to comment more on how the model compares to the pre-existing tools, why differences might exist, how the tool might be used (clinically and/or for population at large). 5. While the risk of PJI after NSAID and anti-fungal use is interesting, what is more interesting is that this relationship was not seen with intra-articular steroid injection or anticoagulants, which does not align with at least some other studies, and raises a question about the accuracy of this relationship. The plausibility of these relationships should be addressed in the discussion. Minor concerns 1. Intro line 69. Do the authors mean after revision arthroplasty, or only revision arthroplasty following prior PJI? The PJI rate is substantially higher for revision procedures even when the revision is not due to prior PJI (and some studies put the risk after prior PJI to be as high as 10-20 times higher). The authors should clarify this, or limit the discussion to revision procedures. 2. Results line 209. The probability of not developing PJI (do the authors mean to include NOT?)
--	---

REVIEWER	Adrien Lemaigen Centre Hospitalier Régional Universitaire de Tours, Service de Médecine Interne et Maladies Infectieuses
REVIEW RETURNED	15-Feb-2024

GENERAL COMMENTS	I read with attention the manuscript by Perni and Prokopovich. Based on national electronic health records, they determined a risk prediction tool for preventing PJI. Major comments: Methods: There are few mentions of bias limitation and this aspect could be reinforced Results: more than 10% of patients did not complete the follow-up. With an incidence of PJI lower than reported in other studies, this fact is important: what is the exhaustivity of PJI detection ? what is the median duration of follow-up for those patients lost-to-follow-up ? Which is the influence on incidence results ? Considering risk equations, authors choose pre-surgical variables in order to make a risk assessment before surgery. In this context, one could be surprised by the inclusion of patella resurfacing in the variables included in the multivariate analysis.
---

	Some intra- or immediate postoperative factors have been described as risk factors for PJI, as duration of surgery, or post-surgical complications (hematoma, second look etc.). It could be interesting to have information on the rate of post-operative immediate complications, which could be associated with preoperative risk factors (i.e. DOAC and hematoma) and being themselves risk factors for PJI. Did the authors have these information in CPRDs ? Discussion: medicoadministrative databases are extremely important in epidemiology, but interpretation of the results is difficult and needs to take into account the uncertainty linked to data quality as they are not initially designed for research purpose Quality assessment of data in CPRD should be discussed. Did the authors or previous works have verified the quality of information reported in CPRD? Notably, no missing data is reported for medical history, which appears suspicious: absence of mention in CPRD could be considered as absence of event and not as a missing data, but this should be specified, and controlled in the source. There is no mention in the authorship or in the acknowledgements of the participation of healthworkers or clinicians implicated in the implementation of CPRDs: this could help to adress these limitations Most results are classical, but some could be discussed: How the authors explain the reduced risk of PJI with ageing ? survival bias ? Osteoporosis appears as a protective factor: adjustment bias with gender and age ? Absence of significant risk for active smoking: is this information well documented in CPRD ? Table S3 is not clear, as the total number of patients (non-smokers + current smokers + ex-smokers) exceeds the total number os subjects. Which variable was used in the equation ? More generally, it is not clear to me how missing data has been handled: missing data are sometimes imputed, and sometimes considered as independant outcome (variable Primary arthroplasty, admission type etc.) Authors could specify which specific treatment of missing data has been used for each variable as the amount of missing data and lost of follow up exceeds the number of events. The potential impact on the results should be discussed in order to ponderate conclusions. Considering the perspectives of these results, most of risk factors are non-modifiable with a limited impact of preventive measures outside of drug choice (but intimely linked with medical history) how the authors imagine the practical use of the risk prediction tool in clinical practice ? Is there a project of external validation of the model in a prospective cohort ? Minor comments: Results section: I.209 : "The probability of not developping PJI" should be "The probability of developping PJI" Risk equations paragraph: A developpment of abbreviations at first occurrence could make the reading easier
--	--

VERSION 1 – AUTHOR RESPONSE

Reviewer: 1

Dr. Sandra Nelson, Massachusetts General Hospital

Comments to the Author:

For the editor:

I did not realize when I accepted this review that the needs of the reviewer would be very heavily weighted towards an understanding of tool development and validation. These are not within my skill set. While I am able to comment on how this study fits into the available literature, I am not sufficiently able to evaluate it on methodologic grounds, and would certainly ensure that there is appropriate evaluation from a statistician or methodologist. Additionally, please note that I am a US physician and my understanding of care systems in the UK is more limited.

General Comments

In this study, the authors sought to develop a risk prediction tool for the development of periprosthetic joint infection following arthroplasty. Several such tools are in existence already but the available tools have primarily been developed in non-UK populations. There are important differences in both population comorbidity risk as well as surgical methods such that development of a tool that is more directly relevant to the UK population is a very reasonable endeavor. The authors used two publicly available data sets to develop the risk prediction tool. Despite differences in the available data sets, the authors were able to demonstrate the reproducibility of the prediction tool which may lead to targeted risk mitigation strategies.

Important issues

1. The reported PJI risk identified in both databases was low (0.4% and 0.5%). For a dataset that is meant to include both early and late PJI, I question whether there may have been missed outcomes in the dataset, as most studies put the true incidence as higher. The authors should address this in their discussion, and whether or not this might introduce bias.

We have added the following the discussion section of our manuscript:

“The observed PJI incidence in this study (~0.7% after 5 years) is lower than that sometimes reported. One possible explanation for this, was the use of a specific ICD-10 code for the identification of PJI (T84.5) while in our studies reported higher PJI rates (about 2.5% over 5 years) additional ICD-10 codes were employed to identify PJI, for example T81.4 (Infection following a procedure) T84.7 (Infection and inflammatory reaction due to other internal orthopedic prosthetic devices, implants and grafts) (Weinstein et al. 2023); however such codes appear less specific and, for example, T81.4 may lead to consider a superficial infection of the surgical wound as a PJI while M00 corresponds to a diagnosis of pyogenic arthritis (Bülow et al. 2022). When using only the presence of ICD-10 code T84.5 the PJI incidence at 5 years observed in this study is similar the 0.89% previously reported (McMaster Arthroplasty Collaborative 2020)”

Unfortunately, as the CPRD protocol did not include other codes than T84.5 for the identification of PJI, we are not able to re-run the analysis to capture these events.

Weinstein EJ, Stephens-Shields AJ, Newcomb CW, et al. Incidence, Microbiological Studies, and Factors Associated With Prosthetic Joint Infection After Total Knee Arthroplasty. JAMA Netw Open.2023;6(10):e2340457

2. The authors included a cohort who underwent their index procedure as long ago as 2007. There have been important improvements in preventive measures over this time that reduced PJI risk (e.g. Staph aureus testing and decolonization). The authors should consider including the time of the index procedure as a covariate.

We thank the review for suggesting the addition of the index date year as covariate; we have rerun our analyses including this and the results are presented in the revised version of the manuscript. We also would like to point out that, despite the surgical methodological advances during the study period, the year of surgery was not retained among the model covariates after step-wise selection.

We also added the following in the discussion section of the manuscript:

“Because of the long follow-up required to capture the full extent of PJI, we chose a study period of 5 years and to allow patients to achieve such observation we restricted the last date of index surgery to 5 years before the last data collection in CPRD extraction analysed”

3. The authors extracted data in December 2020 (methods line 115) and included individuals who underwent index arthroplasty as late as July 2020, yet they also stated that patients were followed up for 5 years or until the first occurrence of the outcome (methods line 130). These are internally inconsistent, and according to table S2, a significant minority of persons in both databases did not complete the 5 year followup. While the data was censored for these individuals, this may still bias the prediction tool towards risk factors associated with early PJI. The authors might consider separate prediction tools based on time following arthroplasty (e.g. early risk – within one year, and later risk, beyond one year after index procedure) or consider this as a sensitivity analysis. If this is not done, the authors should address this in the discussion section.

We took the reviewer suggestion and limited the inclusion to surgeries before 2015 to allow a potential 5 years follow-up. The patient flow (Table S1) and report of censoring events (Table S2) have been updated accordingly.

We thoughtfully considered the possibility of performing a sensitivity analysis targeting specifically early PJI; however, the time provided for resubmission did not allow a full execution of this additional piece of research.

4. The discussion does not really address how this study fits into the available literature. Much of it reads more to me like a supplemental methods section. I would like the authors to comment more on how the model compares to the pre-existing tools, why differences might exist, how the tool might be used (clinically and/or for population at large).

We added the following paragraph in the discussion highlighting the difference of our model against other already published:

“Compared to other available models, beside the UK specific population, our model includes patients both with high BMI (generally considered more at risk of PJI) or unknown BMI, a previous risk scoring model excluded patients with either missing BMI information or BMI > 50 (Bülow et al. 2022). The model developed here considered a wider range of covariates compared to the Mayo Prosthetic Joint Infection Risk Score (Berbari et al. 2012) that included

only diabetes or a limited number (Tan et al. 2018). Moreover, the model presented here is not specific for specific pathogen in contrast with a Staphylococcus aureus specific model (Espindola et al. 2022); this particular relevant in light of the multitude of pathogens isolated in PJI such Gram+, Gram- and fungi (Perni et al. 2023). Additionally, the risk equation presented here has also consideration of a patient drug history that was very limited, if at all, in the other published models (Berbari et al. 2012, Bülow et al. 2022, Espindola et al. 2022, Tan et al. 2018)”

5. While the risk of PJI after NSAID and anti-fungal use is interesting, what is more interesting is that this relationship was not seen with intra-articular steroid injection or anticoagulants, which does not align with at least some other studies, and raises a question about the accuracy of this relationship. The plausibility of these relationships should be addressed in the discussion.

The following has been added in the discussion section of the manuscript:

“The plausibility of the role of NSAID in PJI could be linked the increased cardiovascular risk resulting from their use (Varga et al. 2017) that in turn increases the risk of PJI, it is also widely accepted the immunosuppression is a general risk for the development of PJI and such NSAID role in PJI can also be linked to the immunological activity (Cho et al. 2007). The models did not identify the use of steroidal injection in the joint as a risk associated to PJI in line with the advice of American Academy of Orthopaedic Surgeons (American Academy of Orthopaedic Surgeons 2019). The use of antifungal drug prior to arthroplasty has not been generally investigated as risk factor for PJI; it is possible that a prescription of antifungal drugs before the joint replacement surgery is a risk factor in our model as this would indicate a patient more susceptible to infections”

Minor concerns

1. Intro line 69. Do the authors mean after revision arthroplasty, or only revision arthroplasty following prior PJI? The PJI rate is substantially higher for revision procedures even when the revision is not due to prior PJI (and some studies put the risk after prior PJI to be as high as 10-20 times higher). The authors should clarify this, or limit the discussion to revision procedures.

The reviewer is correct and we have added a clarification, the paragraph now reads:

“...have an incidence of about 1% after primary arthroplasty and about 5 times higher after secondary arthroplasty not resulting from a previous PJI while about 10 time after a revision caused by a PJI.”

2. Results line 209. The probability of not developing PJI (do the authors mean to include NOT?)
We thank the reviewer for highlighting the inconsistency that we now have corrected.

Reviewer: 2

Dr. Adrien Lemaigen, Centre Hospitalier Régional Universitaire de Tours, Université de Tours

Comments to the Author:

I read with attention the manuscript by Perni and Prokopovich. Based on national electronic health records, they determined a risk prediction tool for preventing PJI.

Major comments:

Methods:

There are few mentions of bias limitation and this aspect could be reinforced

Results:

more than 10% of patients did not complete the follow-up. With an incidence of PJI lower than reported in other studies, this fact is important: what is the exhaustivity of PJI detection ? what is the median duration of follow-up for those patients lost-to-follow-up ? Which is the influence on incidence results?

The analysis has been re-run including only patients that had the opportunity of 5 years follow-up (surgery before 2015).

As responded to the previous reviewer the following has been added to the discussion to address the lower incidence that other reports:

“The observed PJI incidence in this study (~0.7% after 5 years) is lower than that sometimes reported. One possible explanation for this, was the use of a specific ICD-10 code for the identification of PJI (T84.5) while in our studies reported higher PJI rates (about 2.5% over 5 years) additional ICD-10 codes were employed to identify PJI, for example T81.4 (Infection following a procedure) T84.7 (Infection and inflammatory reaction due to other internal orthopedic prosthetic devices, implants and grafts) (Weinstein et al. 2023); however such codes appear less specific and, for example, T81.4 may lead to consider a superficial infection of the surgical wound as a PJI. while M00 corresponds to a diagnosis of pyogenic arthritis (Bülow et al. 2022). When using only the presence of ICD-10 code T84.5 the PJI incidence at 5 years observed in this study is similar the 0.89% previously reported (McMaster Arthroplasty Collaborative 2020)”

Unfortunately, as the CPRD protocol did not include other codes than T84.5 for the identification of PJI, we are not able to re-run the analysis to capture these events.

Moreover, the duration of follow-up (mediana, IQR, ...) in patients not completing the study is now presented in Table S3.

Considering risk equations, authors choose pre-surgical variables in order to make a risk assessment before surgery. In this context, one could be surprised by the inclusion of patella resurfacing in the variables included in the multivariate analysis.

We included patella resurfacing among other variables related to the surgery such as the fixation method or the laterality of the operation.

Some intra- or immediate postoperative factors have been described as risk factors for PJI, as duration of surgery, or post-surgical complications (hematoma, second look etc.). It could be interesting to have information on the rate of post-operative immediate complications, which could be associated with preoperative risk factors (i.e. DOAC and hematoma) and being themselves risk factors for PJI. Did the authors have these information in CPRDs ?

The reviewer raises an interesting point that we wanted to consider as well; unfortunately many information related to the surgery are not available in the CPRD/HES database (such as surgery duration, only date) or very incomplete (such as type of anaesthesia).

Moreover, we used only the ICD-10 code T84.5 so we did not capture post-operative immediate complications in our reported incidence. The reviewer suggestion is interesting (early complications as possible risk factor for later PJI) unfortunately, as the CPRD protocol did not include other codes than T84.5 for the identification of PJI, we are not able to re-run the analysis to capture these events.

Discussion:

medicoadministrative databases are extremely important in epidemiology, but interpretation of the results is difficult and needs to take into account the uncertainty linked to data quality as they are not initially designed for research purpose

Quality assessment of data in CPRD should be discussed. Did the authors or previous works have verified the quality of information reported in CPRD?

We added the following to the introduction section to support the data quality concern:

“CPRD GOLD and AURUM database have been shown to be a data source with quality and completeness (Jick et al. 2023)”

Jick S, Vasilakis-Scaramozza C, Persson R, Neasham D, Kafatos G, Hagberg KW. Use of the CPRD Aurum Database: Insights Gained from New Data Quality Assessments. Clin Epidemiol. 2023;15:1219-1222

Notably, no missing data is reported for medical history, which appears suspicious: absence of mention in CPRD could be considered as absence of event and not as a missing data, but this should be specified, and controlled in the source.

There is no mention in the authorship or in the acknowledgement of the participation of health workers or clinicians implicated in the implementation of CPRDs: this could help to address these limitations.

The reviewer is correct and we have specified the following in the material and methods where a section for missing data handling was added:

“Medical history was based on the presence of a specific diagnostic code, therefore the absence of such codes in a patient records was assumed to represent absence of such event, therefore no missing data were possible in a patient medical history. On the contrary, when a record was expected, such in case of BMI, age, primary or secondary arthroplasty along with alcohol and smoking status, the absence of any entry resulted in this patient having the

covariate under consideration categorized as missing. Missing data were handled through categorical binning and no imputation methods applied”

It is not possible to query CPRD for these instances as this was not a primary data collection.

Most results are classical, but some could be discussed:

How the authors explain the reduced risk of PJI with ageing ? survival bias ?

The following text has been added in the discussion section:

“Age is sometimes not reported as a risk factor for PJI (Inoue et al. 2019, Kunutsor et al. 2017), however the protective role of aging on PJI observed in this study was also seen previously. (McMaster Arthroplasty Collaborative, 2020) A possible explanation for such correlation is “survival bias” as older patients dying at higher rates would be censored at the time of death before having the opportunity of developing PJI”

Osteoporosis appears as a protective factor: adjustment bias with gender and age ?

Osteoporosis is not a risk factor anymore after re-running the analyses limiting to surgery before 2015 so no action was taken relating to this comment.

Absence of significant risk for active smoking: is this information well documented in CPRD ? Table S3 is not clear, as the total number of patients (non-smokers + current smokers + ex-smokers) exceeds the total number of subjects. Which variable was used in the equation ?

We have re-run the analyses and the data reported in Tables S3 (*now Table S4) have been checked and the number of patients in each category adds up to the expected total number of patients. We thank the reviewer for spotting the error (a row had been delated). Based on our observation <10% of patients had an unknown smoking status so it is appears this information is relatively well reported in both CPRD databases.

More generally, it is not clear to me how missing data has been handled: missing data are sometimes imputed, and sometimes considered as independent outcome (variable Primary arthroplasty, admission type etc.)

See response to previous comment.

Authors could specify which specific treatment of missing data has been used for each variable as the amount of missing data and lost of follow up exceeds the number of events.

We stated the following in the material and methods where a section for missing data handling was added:

“Medical history was based on the presence of a specific diagnostic code, therefore the absence of such codes in a patient records was assumed to represent absence of such event, therefore no missing data were possible in a patient medical history. On the contrary, when a record was expected, such in case of BMI, age, primary or secondary arthroplasty along with alcohol and smoking status, the absence of any entry resulted in this patient having the covariate under consideration categorized as missing. Missing data were handled through categorical binning and no imputation methods applied”

It is not possible to query CPRD for these instances as this was not a primary data collection.

The potential impact on the results should be discussed in order to ponderate conclusions.

We added the following at the end of the discussion section:

“Missing data have been handled through categorical binning and no mechanisms related to missingness have been hypothesised (i.e. missing at random or missing not at random); in order to avoid selection bias, the analysis has not been restricted to complete cases. Sensitivity analysis and multiple imputations could be further applied to refine the risk scoring algorithm presented”

Considering the perspectives of these results, most of risk factors are non-modifiable with a limited impact of preventive measures outside of drug choice (but in timely linked with medical history) how the authors imagine the practical use of the risk prediction tool in clinical practice ?

The following has been added in the discussion section:

“As most of the identified risks are not modifiable the application of this risk scoring algorithm would not primarily allowing a personalized pre-surgery drug prescription consideration but will guide and follow-up care pivoting the resources (visits, test etc..) to the subgroup of patients at higher risks of PJI”

Is there a project of external validation of the model in a prospective cohort ?

Funding bodies and potential partners are currently being approached in order to design and implement a prospective study as rightly suggested by the reviewer.

Minor comments:

Results section:

I.209 : "The probability of not developing PJI" should be "The probability of developing PJI"
We thank the reviewer for highlighting the inconsistency that we now have corrected.

Risk equations paragraph: A development of abbreviations at first occurrence could make the reading easier

The abbreviations have been now spelt out at the first time of use.